# Elevation and Human Disturbance Interactively Influence the Patterns of Insect Diversity on the Southeastern Periphery of the Tibetan Plateau

**DOI:** 10.3390/insects15090669

**Published:** 2024-09-04

**Authors:** Zhouyang Liao, Jinlu Zhang, Xuemei Shen, Mi Zhu, Xinlin Lan, Junming Cui, Yunfang Guan, Ying Zhang, Zhongjian Deng, Tiantian Tang, Fang Liu, Darong Yang, Yuan Zhang

**Affiliations:** 1Yunnan Academy of Biodiversity, Southwest Forestry University, Kunming 650224, China; zhouyang_liao@swfu.edu.cn (Z.L.); zj_deng@swfu.edu.cn (Z.D.); tiantiantang@swfu.edu.cn (T.T.); 2Institute of Zoology, Guangdong Academy of Sciences, Guangzhou 510260, China; 3Xishuangbanna Tropical Botanical Garden, Chinese Academy of Sciences, Mengla 666303, China; 4Key Laboratory of Southwest Mountain Forest Resources Conservation and Utilization, Ministry of Education, Kunming 650224, China

**Keywords:** elevation, human disturbance, interaction, insect diversity, southeastern periphery of the Tibetan Plateau, redundancy analysis

## Abstract

**Simple Summary:**

In the face of global climate change and escalating human disturbances, insect diversity is facing significant threats. This study aims to investigate the impacts of altitude and human disturbances, as well as their interactions, on insect diversity to establish a strong scientific foundation for effective management measures. The study site selected is located on the southeastern margin of the Tibetan Plateau, one of the world’s prominent biodiversity hotspots. We believe that insect diversity plays an important role in the maintenance of biodiversity in this region. A sufficient survey of insect diversity in the area, analyzed in relation to elevation and human disturbances, revealed that insect diversity declined with increasing elevation, moderate disturbance promoted insect diversity to a certain extent, and heavy disturbances significantly reduced it, with an interaction between elevation and human disturbance. Further analyses revealed that the effects of human disturbance on insect diversity varied across elevation. The insect diversity below ~2200 m elevation was higher under low disturbance compared to moderate and heavy disturbance, while above ~2200 m elevation, it was higher under moderate disturbance than low and heavy disturbance. Additionally, different insect taxa showed varying responses to the combination of elevation and disturbance. In this study, the majority of insect taxa exhibited a higher diversity at lower elevations and under low to moderate disturbance conditions, while Hymenoptera demonstrated their ability to maintain high diversity even in areas with high elevation and human disturbances. The findings suggest that when implementing integrated insect management, it is crucial to consider the impacts of elevation and human disturbances on specific insect taxa. Our results not only enhance our comprehension of the factors influencing the diversity and distribution of insects, but also establish a scientific foundation for formulating effective biodiversity conservation and management programs.

**Abstract:**

The maintenance of biodiversity and ecological balance heavily relies on the diversity of insects. In order to investigate the impacts of elevation and human disturbance, as well as their interactions on insect diversity, we conducted an intensive survey of insects in the Hengduan Mountain Range, which is situated on the southeastern periphery of the Tibetan Plateau in China. A total of 50 line transects were established in this study to investigate the impact of elevation and human disturbance on insect diversity and distribution patterns. Designed insect surveys were conducted at various elevations and levels of human disturbance, and statistical methods such as generalized linear modeling and redundancy analysis were employed for data analysis. The results of this study indicated a negative correlation between insect diversity and elevation. Additionally, moderate disturbance was found to have a positive impact on insect diversity to some extent. The explanatory power of the model for the distribution of insect diversity could be improved if elevation and human disturbance were included as an interaction effect into the model, and there were differences in the effects of human disturbances on insect diversity at different elevation levels. The highest insect diversity was observed under low disturbance conditions below elevation of ~2200 m, whereas above this threshold, insect diversity was the highest under moderate disturbance compared to low disturbance. The response of different insect taxa to the interactions of elevation and human disturbance varied. The findings imply that when formulating strategies for managing insect diversity, it is crucial to thoroughly consider the interaction of environmental factors and disturbance response of individual insect taxa.

## 1. Introduction

With the rapid decline of global biodiversity, biodiversity conservation has become one of the most urgent global issues [1,2,3]. As one of the most abundant biological groups, insects play a crucial role in maintaining ecological stability in nature [4]. Different insect groups play different important roles in ecosystem services, acting as essential pollinators, natural enemies, decomposers, etc. [5,6]. Furthermore, certain insect groups can be used as biological indicators for assessing environmental quality, as their community dynamics can reflect ecosystem condition [7]. The spatial distribution patterns of insects are also regarded as comprehensive indicators of dynamic changes in ecosystems. However, insect diversity is under serious threat due to various factors, including the impacts of environmental and climate changes [8,9], which is of growing concern to entomologists.

Elevation gradients, which are a crucial component of the natural environment, may significantly impact insect diversity [10]. Temperature and other climatic conditions often change with elevation, and directly affect the survival and reproduction of insects [11]. For instance, low temperatures in high-elevation areas hinder the development of some insect species, thereby reducing the diversity of insects in these regions [12,13]. Furthermore, the uniqueness of ecosystems in high-altitude areas may result in significantly different insect community structures compared to low-altitude regions [14]. Additionally, global temperature increases may also lead to the migration of insects from lower to higher elevations, thus altering the community structure of insects in these areas [15]. Overall, insect diversity may exhibit certain patterns with altitude, but distinct differences in the effects of altitude changes may exist among different taxa [16,17,18,19]. Previous studies have shown that species richness may exhibit a unimodal or decreasing pattern with increasing elevation [20,21], while other studies have shown that there may be a bimodal or increasing pattern between species richness and elevation [22,23]. Currently, studies on the correlation between elevation and insect diversity have not reached a consensus, and most studies have focused on a single or only a few taxa.

In addition to elevation, the rapid increase of human activities over the past few decades has also made human disturbance one of the main causes of global insect diversity loss [24]. Human disturbances, including agricultural development, urbanization, deforestation, etc., have altered the natural habitats of insects, exerting both direct and indirect effects on insect communities [25]. For example, habitat fragmentation caused by human disturbances has restricted the migration and dispersion of insect populations, thereby reducing species diversity [24]. Moreover, using pesticides and fertilizers in agricultural activities severely affects the survival and reproduction of insects [26]. Human disturbances may affect insect diversity at the levels of species, populations, and communities [27,28]. On the one hand, frequent human disturbances prevent some species from adapting to new environments by altering environmental conditions, thereby affecting insect population structures [29]. On the other hand, disturbances may lead to changes in food resources for insects, thereby limiting the numbers of certain species in specific areas [30]. Moreover, human disturbances may also facilitate the invasion of alien species, increasing competitive pressures on native species and further reducing biodiversity [31]. Meanwhile, some studies have indicated that moderate human disturbances may promote insect diversity. For example, shrubbery and semi-natural grasslands at the edges of farmland may contribute to increasing the species richness of insects [32]. This phenomenon can be explained by *the Intermediate Disturbance Hypothesis* (IDH), which posits that moderate disturbances in an ecosystem can promote environmental heterogeneity, thereby promoting coexistence among more species [33]. Furthermore, current research on insect diversity often focuses on specific species or particular insect groups, failing to adequately reflect the overall complexity of insect taxa [34,35].

Currently, the impacts of elevation and human disturbance on biodiversity have become important areas in biodiversity research, yet most existing research typically considers these two factors independently [36,37]. Studies on other biological groups imply that elevation and human disturbance may interact, thus potentially having more complex effects on biodiversity than when considered as single factors [38]. However, few papers have focused on the effects of these two combined factors on insect diversity and their potential interactions. Changes in elevation often coincide with alterations in environmental conditions, encompassing crucial factors such as temperature, humidity, vegetation type, and resource availability that directly impact insect diversity [39]. However, human disturbances such as farmland reclamation, deforestation, and urban expansion can significantly impact environmental heterogeneity by either creating new habitats or destroying pre-existing ones, thereby exerting an influence on insect diversity [27,29]. Additionally, the same level of human disturbance may have different impacts on insect diversity at different elevations. Therefore, including both elevation and human disturbance in studies not only helps us better understand the distribution patterns of insects, but is also crucial for predicting and understanding future trends in biodiversity under the context of global changes [40,41].

Based on the above considerations, we carried out a survey and study of insect diversity in the Hengduan Mountains, which is locate at the Sichuan–Tibet–Yunnan junction in China, on the southeastern periphery of the Tibetan Plateau. This region is the longest, widest, and most typical north–south mountain range in China, and it is one of the world’s youngest mountain ranges, as well as one of the 36 global biodiversity hotspots [42]. We believe that insect diversity plays an important role in the maintenance of biodiversity in this area, but due to factors such as transport and climate condition, few surveys of insect diversity in this area, as well as research work, have been carried out. We planned to investigate insect diversity within elevations ranging from 1600 to 3800 m and conduct in-depth analyses of insect community structures, as well as the distribution patterns of specific insect groups at different elevations and levels of human disturbance. Data analysis methods such as generalized linear models and redundancy analysis were also used to explore the effects of these two factors on insect diversity and distribution. The results of this study will help us to understand the distribution of insect diversity in the region more comprehensively and provide a scientific reference for us to further explore the interaction between altitude and human disturbance.

## 2. Materials and Methods

### 2.1. Study Area

The study site is located in northwest Yunnan, China. It is situated on the southeastern periphery of the Tibetan Plateau, spanning between 98°35′ and 100°19′ E longitude and 26°52′ and 29°16′ N latitude. The total area it covers is approximately to 23,870 square kilometers. The region encompasses an altitude range of 5254 m, stretching from the Lancang River basin’s lowest point at 1486 m to Kawagabo Peak on Meili Snow Mountain, which stands at a height of 6740 m. The study area encompasses three distinct vertical ecological environments: the valley region (1486–2200 m), the mountain area (2200–2800 m), and the alpine area (2800–6740 m). The temperature fluctuates between −27.4 °C and 25.1 °C, with an average annual temperature ranging from 4.7 °C to 16.5 °C. The average annual sunshine duration ranges from 1740 to 2190 h, while the yearly rainfall varies between 600 and 1000 mm. The unique geographical location, diverse terrain, and varied climatic conditions contribute to making it one of the most biologically diverse areas in the world [43].

### 2.2. Study Design

The objective of this study was to examine the combined impacts of elevation and human disturbance on insect community structure. The design of the study sites takes into consideration both scientific norms and accessibility, and was arranged along elevation gradients. This study involved the establishment of 50 line transects, each measuring 1 km in length and 5 m in width. The latitude, longitude, and elevation for each line transect were meticulously recorded using GPS devices. The research area was divided into five distinct elevation gradients based on the diverse range of elevations and vertical habitat types in the survey region, as well as preliminary survey findings. The elevation range and number of line transects for each gradient were as follows: gradient 1 (1600–1900 m) with 13 lines, gradient 2 (1901–2200 m) with 13 lines, gradient 3 (2201–2500 m) with 10 lines, gradient 4 (2501–2800 m) with 7 lines, and gradient 5 (2801–3800 m) with 7 lines (Figure 1). The intensity of human disturbance in each line transect was classified into three distinct levels: low, moderate, and high, based on the *Integrated Disturbance Index* (IDI) [44]. IDI values less than 0.30 were considered as low-disturbed; IDI values more than 0.65 were considered as high-disturbed; IDI values between these thresholds were considered as moderately-disturbed [36]. All disturbance intensities were designed under each elevation gradient to minimize the general collinearity between elevation and human disturbance.

### 2.3. Insects Sampling and Species Identification

Based on the climatic conditions and seasonal characteristics of the insect community in the survey region, a preliminary investigation was conducted, revealing that there is virtually no insect activity during winter and spring in the middle and high-elevation areas of the investigation sites, making it impractical to collect insects. Subsequently, survey sites were designated for four insect surveys, and the surveys were carried out during the summer and fall seasons of 2021 and 2022, encompassing a total of 50 line transects each time. The survey of each line was conducted by a team of two investigators. The surveys were conducted using the sweeping method, with a speed of 1–2 km per hour along each line, covering a range of 5 m for insect collection around the line. The captured insect samples, categorized by taxa, were preserved in 75% alcohol-filled plastic bottles or triangular paper envelopes and transported to the laboratory for specimen preparation and identification. In consideration of the complexity of insect taxonomy and the prevalence of unnamed insect species, we employed the family as the taxonomic unit and morphospecies for classifying the collected insects. The effectiveness and rationality of this approach have been verified and successfully applied in many studies on insect diversity [45]. The taxonomic identification of all collected insects was conducted by insect taxonomists, who referred to relevant specialized books and references [46,47,48,49,50].

### 2.4. Data Analysis

The data analysis and graphing in this study were conducted using R version 4.2.2. The adequacy of the insect samples was assessed using the “iNEXT” package [51]. This study utilized the “vegan” package to calculate the insect diversity indices (including the richness, Shannon–Wiener, and Simpson indices) for each sampling site [52], aiming to investigate the impacts of elevation gradient and human disturbance intensity on insect Alpha diversity. We also computed the Chao index, which serves as a predictive measure for estimating the total richness within an indigenous insect community. The Chao index provides an estimation of the number of unobserved species, thereby enhancing the overall comprehension of biodiversity [53]. The differences of the insect diversity indices were assessed using a one-way ANOVA analysis and the subsequent Fisher’s Least Significant Difference (LSD) method. The relationship between insect species abundance and elevation, as well as disturbance gradients, was thoroughly investigated using a generalized linear model (GLM).

By analyzing four potential hypothetical models, a more comprehensive understanding of the impacts of various factors on insect diversity was attained [54]. The models include: (i) elevation model—elevation independently accounts for variations in insect diversity patterns; (ii) human disturbance model—human disturbance independently accounts for variations in insect diversity patterns; (iii) elevation + human disturbance (additive) model—simple additive effects of elevation and human disturbance account for variations in insect diversity patterns; and (iv) elevation × human disturbance (interaction) model—the interaction of elevation and human disturbance account for variations in insect diversity patterns. The R^2^ values and *Akaike Information Criterion* (AIC) values were also computed for each model to assess the fitting degree [55]. The R package “MuMIn” was employed to perform model averaging for models with ΔAIC values less than 2, aiming to enhance the accuracy of the results [56]. The R package “car” was employed to assess collinearity among the variables in our model. The analysis resulted in the identification of 15 groups, which were derived from five elevation gradients and three human disturbance gradients. The correlation between these groups and those of different insect taxa was investigated using a thermal image tool based on the “pheatmap” package [57]. The effects of elevation and human disturbance on insect diversity, and the similarity of insect communities across elevation and disturbance gradients were examined using *redundancy analysis* (RDA) based on the “vegan” package [52]. The results were deemed statistically significant at a level of *p* < 0.05.

## 3. Results

### 3.1. The Assessment of Species Composition and Sampling Adequacy

A total of 5141 insects belonging to 13 orders and 80 families were collected in this study (Table 1 and Appendix A). The insect communities were predominantly composed of Coleoptera and Hemiptera, with other significant orders including Hymenoptera and Orthoptera. Diptera also constituted a notable proportion of the families present, although with relatively lower individual abundance. The diversity of families and individuals in the other insect orders was relatively limited in this study, however, it still demonstrated a certain level of variation. The sampling adequacy test, based on the sample size, indicated that the survey data effectively represented the insect diversity within the studied area (Figure 2).

### 3.2. The Independent Impacts of Elevation and Human Disturbance on Insect Diversity

In studies examining changes along elevation gradients, we observed that the insect richness index (df = 4, F = 2.89, *p* < 0.05, Figure 3A) and the Chao index (df = 4, F = 2.61, *p* < 0.05, Figure 3B) initially increased and then decreased with elevation, peaking at the second elevation gradient. The Shannon diversity index (df = 4, F = 3.26, *p* < 0.05, Figure 3C) and the Simpson diversity index (df = 4, F = 3.59, *p* < 0.05, Figure 3D) exhibited negative correlations with increasing elevation. A more pronounced decrease in the Alpha diversity index of insect communities was observed between the second and third elevation gradients compared to adjacent gradients, with significant variations noted across the different elevation gradients. In studies on human disturbance changes, the insect richness index (df = 2, F = 6.11, *p* < 0.05, Figure 4A), the Chao index (df = 2, F = 4.36, *p* < 0.05, Figure 4B), the Shannon diversity index (df = 2, F = 3.56, *p* < 0.05, Figure 4C), and the Simpson diversity index (df = 2, F = 2.01, *p* = 0.14, Figure 4D) all initially increased and then decreased with increasing levels of human disturbance. The Alpha diversity index was the highest at moderate disturbance levels and the lowest under high disturbance conditions. The overall results showed that human disturbance significantly impacted the Alpha diversity of insect communities.

### 3.3. Hypothesis Testing for the Interaction of Elevation and Human Disturbance

In multi-model inference, a lower AIC value indicates a better model performance, as does a higher R^2^ value. Our model calculated both AIC and R^2^ values. The results indicated that the model exhibited a lower level of explanatory power for insect diversity when either elevation or human disturbance was considered as separate dependent variables. However, the efficacy of the model improved when both elevation and human disturbance were included as independent variables. The further inclusion of elevation and human disturbance as interactive factors led to optimal explanatory power for insect diversity (Table 2).

The data visualization results further demonstrated a negative correlation between insect richness and elevation, regardless of the level of disturbance. Below 2200 m in elevation, the richness of insects under low disturbance was higher than that under moderate disturbance. Above 2200 m, richness under moderate disturbance exceeded that under low disturbance. Across all elevation gradients, richness was lowest in habitats with high disturbance (Figure 5).

### 3.4. Responses of Various Insect Taxonomic Groups to the Interaction of Elevation and Human Disturbance

Further heatmap analysis of various insect groups across different elevations and levels of human disturbance revealed significant variations in species composition among the samples (Figure 6). Our results indicated that most insect groups exhibited higher richness under low-elevation conditions with low-to-moderate disturbance. The Hymenoptera, however, exhibited greater species diversity at high elevations under low-to-moderate disturbance. Additionally, certain groups such as Trichoptera exhibited lower richness at most of the sample sites but demonstrated higher richness at specific sites. The results of the cluster analysis revealed that the distribution patterns of insect species were not exclusively determined by elevation or human disturbance. Instead, distinct differences in insect community composition were observed as a result of various combinations of elevation and human disturbance.

### 3.5. The Analysis of Insect Community Similarities across Varying Elevations and Levels of Human Disturbance

Redundancy analysis further demonstrated the impact of elevation and human disturbance on the distribution patterns of insect communities. The similarity of insect communities across different elevation gradients exhibited significant variation, but the distribution pattern of insect communities was not merely based on elevation gradient or human disturbance alone. Conversely, the interplay between elevation and human disturbance exerted a notable impact on the distribution patterns of insect communities. The ranking results obtained from different sample sites exhibited similarity to the clustering results, providing further evidence for the intricate interactions between elevation and human disturbance that influence insect diversity (Figure 7).

## 4. Discussion

Elevation and human disturbances have been demonstrated as significant factors influencing biodiversity in several studies [10,29]. Although some published studies have investigated the impacts of these two factors on insect diversity, the majority of them have solely considered these factors as independent variables or focused on a single insect taxon [15,36]. Consequently, there is a dearth of research examining the interactions between these two factors and limited reports addressing the integration of multiple taxa. In this study, the effects of elevation and human disturbances on insect diversity at the southeastern periphery of the Tibetan Plateau were investigated through adequate systematic investigations and scientific analyses. The results showed that insect diversity declined with the increase in elevation when elevation was taken into account alone. This finding aligns with the findings of a study on Lepidoptera in the Swiss Alps, which also demonstrated a negative correlation between elevation and species richness of Lepidoptera [58]. Studies of some other taxa have also shown that lower temperatures associated with increased elevation and homogenization of habitat types, as well as endemism, can lead to a decrease in the diversity of some biological groups [59,60,61]. However, the pattern of insect community change along with the elevation is still controversial. The relationship between insect diversity and elevation has been observed to exhibit a unimodal pattern in some studies [62,63], while others have demonstrated a bimodal pattern [64,65]. We found that the discrepancies in the findings of these studies were attributed to variations in the taxonomic composition, differences in elevation ranges, or site-specific environmental factors. In addition to considering the elevation factor, we also examined the impact of human disturbance on insect diversity. When solely considering the human disturbance factor, there was an observed trend in insect diversity where it initially increased and then decreased with an increasing intensity of disturbance. Notably, under moderate levels of human disturbance, insect diversity reached its highest value, indicating that moderate disturbances had a positive effect on promoting insect diversity when other factors were not taken into account. This is consistent with the findings of a study on arthropod diversity conducted in a subtropical forest in South Africa [66]. Based on our findings and relevant studies on the diversity of other biological groups, it has been demonstrated that the distribution pattern of biodiversity can be influenced by a combination of various factors [67]. Solely considering a single factor often fails to adequately elucidate the determinants impacting the distribution of biodiversity. In order to comprehensively understand the key factors affecting insect diversity, it is essential that studies on both elevation and human disturbance are included.

In order to explore the general distribution pattern influencing insect diversity, we incorporated all surveyed taxa into our analysis. The findings revealed that there is an interactive effect between elevation and human disturbances in shaping insect diversity. The results of the multi-model inference indicated that the explanatory power of models significantly increased when considering the interaction between elevation and human disturbance compared to solely examining each factor individually. Furthermore, both cluster analysis and redundancy analysis also demonstrated that the interpretive efficacy of insect diversity is diminished when solely considering elevation or human disturbance factors. Additionally, we performed a *variance inflation factor* (VIF) analysis for all models. The VIF value showed that the collinearity between elevation and human disturbance was not significant, and both factors can be considered as independent variables, further validating the reliability of the model. Our findings also indicated considerable variations in the composition and diversity of insect communities across different elevations and disturbance combinations, thereby highlighting the combined influence of these two factors on determining insect community composition. Although overall, lower elevations exhibit greater climate and habitat heterogeneity, providing abundant resources for insects, resulting in a decline in insect diversity with increasing elevation, our findings reveal an intriguing pattern: at low elevations, insect diversity reaches its peak under conditions of low disturbance; however, above ~2200 m elevation, moderate disturbance leads to higher insect diversity compared to low disturbance levels. The results suggest that elevation and human disturbance may have a combined impact on insect diversity through their influence on environmental heterogeneity. Specifically, low-elevation areas may exhibit the highest biodiversity capacity in their original state [68], and moderate disturbances could potentially undermine these favorable habitat conditions. The reduction in insect diversity in higher elevation areas can be attributed to specific habitat conditions, such as hydrothermal characteristics and vegetation types, which are exclusively suitable for particular insect groups [69,70]. Since the original habitats in high-elevation areas are intrinsically not equipped with characteristics for supporting higher biodiversity, moderate disturbances can actually benefit more insect species by creating greater habitat heterogeneity, thus resulting in higher insect diversity [71,72]. For instance, at elevations above ~2200 m in our study area, we have observed that the introduction of some ornamental plants, which would be uncommon at these elevations without factors such as tourism development and garden landscaping, has resulted in the increased attraction of pollinators like Hymenoptera and Diptera, consequently leading to a significant enhancement in insect diversity within the region. However, low disturbances may not be sufficient to alter the original environmental conditions of high-altitude areas, so insect diversity is lower than under moderate disturbance [73]. So, in general, although ecologically, ecosystems at lower elevations have a high resistance to disturbances, such disturbances may still lead to a reduction in insect abundance to some extent [74]. Our findings also indicated that high disturbances were associated with the lowest biodiversity at all elevations. This could be attributed to the extent of habitat destruction caused by high disturbances, surpassing the ecosystem’s resilience or capacity for recovery, thereby severely impacting species survival and reproduction, and exerting a significant negative influence on biodiversity [75]. The findings of our study provide partial support for the moderate interference hypothesis, while also suggesting that certain prerequisites may be necessary for this hypothesis to hold true [76]. Therefore, when applying moderate disturbance theory to guide biodiversity conservation and management, it is necessary to fully consider the specific conditions of the region.

The response of insect diversity to the interaction between elevation and human disturbance exhibited a certain regularity, while further analysis revealed that different insect groups had different responses to the interaction. For instance, in lower elevations with low-to-moderate disturbance levels, taxa such as Hemiptera, Diptera, Orthoptera, and Lepidoptera exhibited greater richness. This suggests that these groups have a preference for habitats with favorable hydrothermal conditions and more stable environments [21,77], and may possess less adaptability to high elevations and highly disturbed environments. We also found that Hymenoptera showed high richness in low-to-moderate disturbances environments across all elevation, and the presence of hymenopteran such as bumblebees can still be observed in the region at an altitude of 3800 m. The findings demonstrate the remarkable adaptability of Hymenoptera to various environmental climates and human disturbances, providing important ecological services to the natural world [78]. Our study demonstrates that the Tibetan Plateau region exhibits rich insect diversity. The responses of different insect taxa to the interaction between elevation and human disturbance varies obviously. Moreover, we found that moderate levels of human disturbance promote an increase in the abundance of specific insect species. These findings suggest that various groups of insects employ distinct adaptive strategies to cope with environmental changes, thereby providing a scientific foundation for understanding the mechanisms underlying the maintenance of insect diversity under diverse environmental stresses [79].

Different insect taxa may exhibit diverse ecological roles in natural ecosystems, making their scientific conservation essential for maintaining ecological stability [5]. This study extensively examines the impacts of elevation and human disturbances, as well as their interactions, on insect diversity in the southeastern periphery of the Qinghai–Tibet Plateau through comprehensive field investigations and scientific data analysis. It represents an exploratory investigation into the intricate relationship between human disturbance, elevation, and biodiversity. Against the backdrop of the intensification of global climate change and the increasing fragmentation of habitats, our study offers a meaningful scientific foundation for insect diversity conservation and management while also serving as a valuable reference for investigating other biological groups. In the future, it is crucial that comprehensive and long-term studies on insect diversity are conducted, particularly in biodiversity hotspots. By expanding the geographical scope, incorporating a broader range of environmental variables, and adapting sampling methods to different insect groups, we can enhance our understanding of the factors influencing insect diversity and distribution. This will provide a more valuable scientific foundation for addressing the challenges posed by global change to biodiversity conservation and management.

## 5. Conclusions

Our results reveal the critical role of the interaction between elevation and human disturbance in determining insect diversity and distribution. The findings demonstrate that both elevation gradients and the intensity of human disturbance exert significant impacts on insect diversity; however, incorporating these two factors as interacting variables into the model substantially enhances its predictive capacity for insect diversity. Our findings suggest that low disturbance is beneficial for maintaining high insect diversity at lower elevations, whereas moderate disturbance is more advantageous for preserving insect diversity at higher elevations. Additionally, further analysis of the distributional characteristics of different taxa revealed distinct responses to the combined effects of disturbance and elevation. The distributional characteristics of specific insect taxa, as well as variations in disturbance resistance, should be thoroughly considered when formulating management strategies related to insect diversity. Clearly, the factors influencing the distribution of insect diversity can be multifaceted, and our study suggests that future research incorporating a broader range of environmental factors is warranted.

## Figures and Tables

**Figure 1 insects-15-00669-f001:**
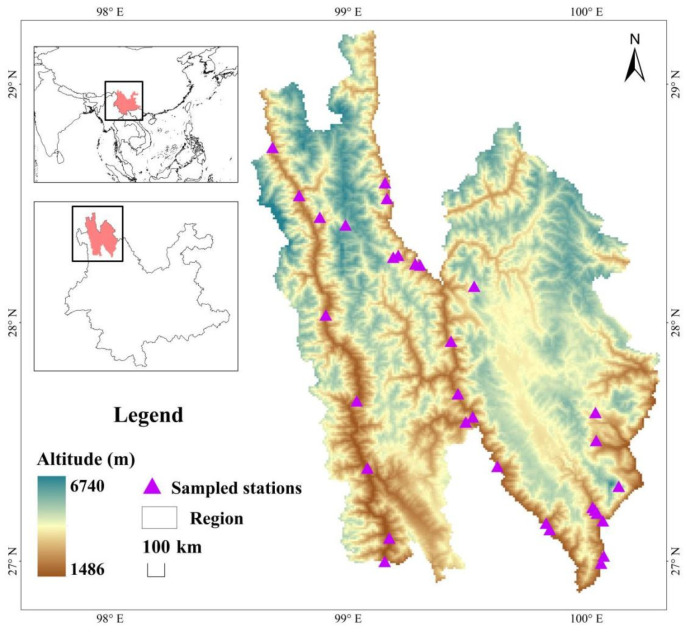
The field investigation area and sampling sites.

**Figure 2 insects-15-00669-f002:**
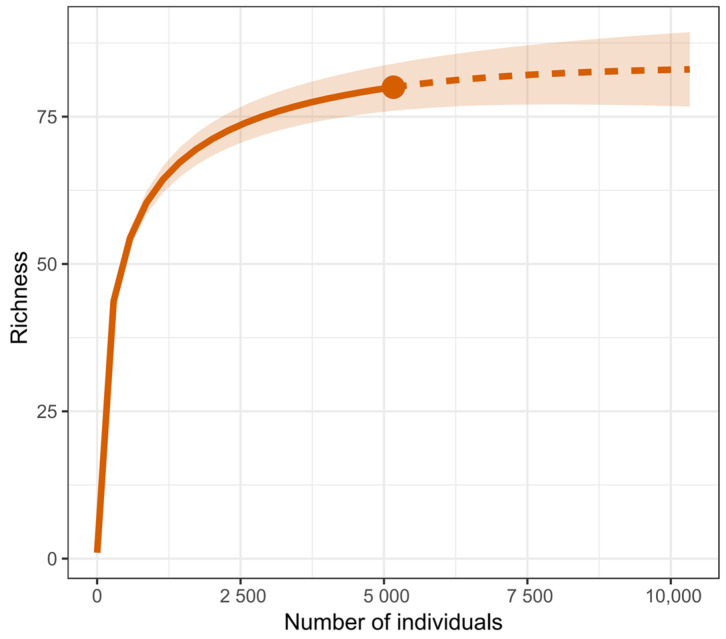
The sampling adequacy test based on insect richness with a confidence interval of 95%.

**Figure 3 insects-15-00669-f003:**
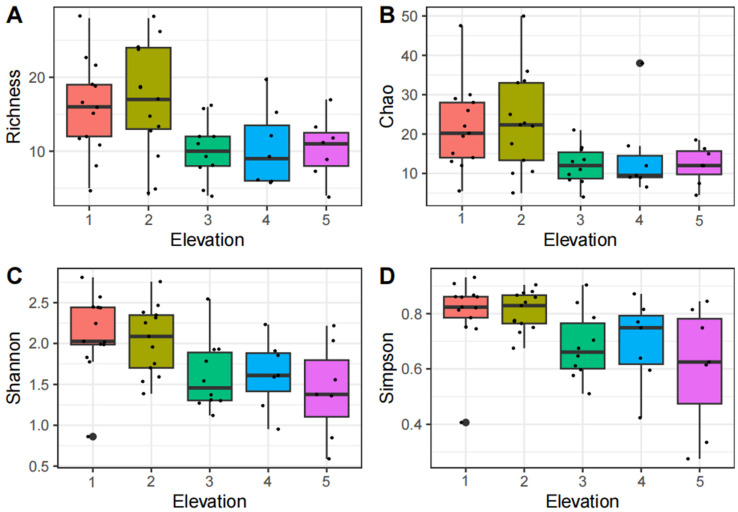
The Alpha diversity of insect communities at different elevation gradients. Note: 1, 2, 3, 4, and 5 represent elevation gradients 1, 2, 3, 4, and 5, respectively (see study design for each elevation range). Richness, Chao, Shannon and Simpson indices are shown in the (**A**,**B**,**C**,**D**) panels, respectively.

**Figure 4 insects-15-00669-f004:**
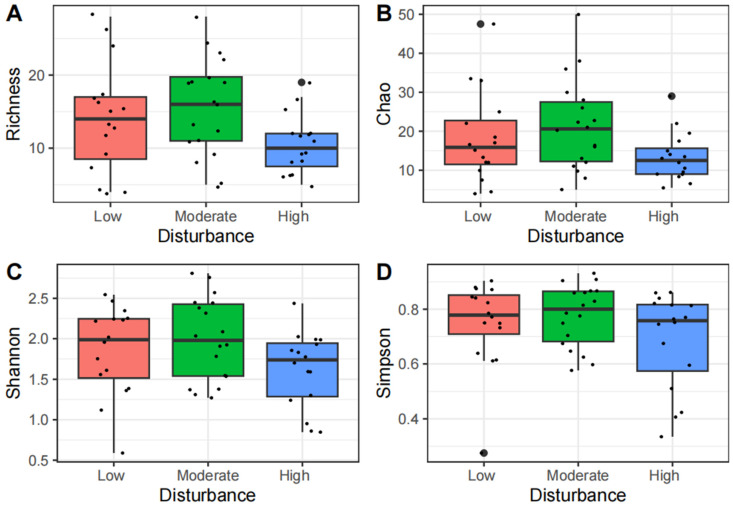
The Alpha diversity of insect communities at different levels of human disturbance. Richness, Chao, Shannon and Simpson indices are shown in the (**A**,**B**,**C**,**D**) panels, respectively.

**Figure 5 insects-15-00669-f005:**
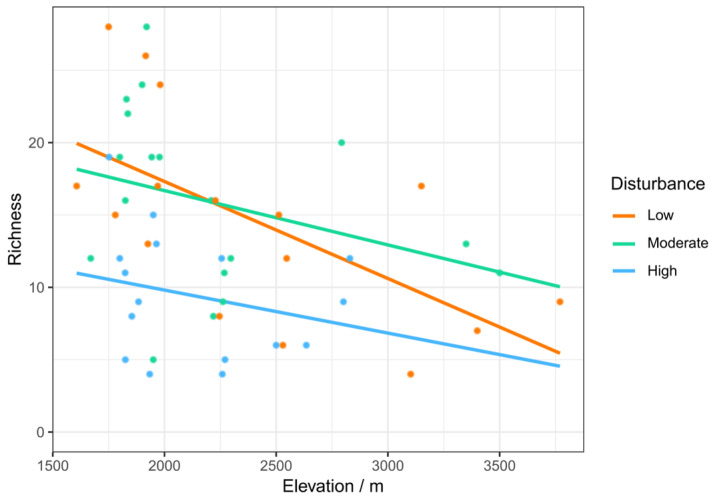
The influence of the interaction between elevation and human disturbance on insect richness.

**Figure 6 insects-15-00669-f006:**
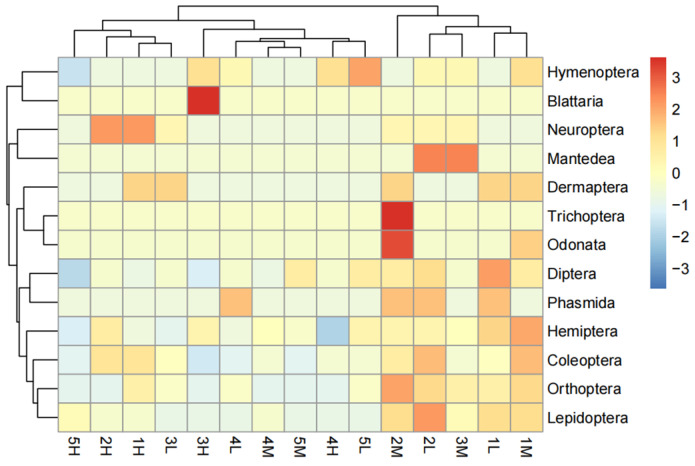
Responses of different insect taxonomic groups to the interaction of elevation and human disturbance. Note: The combination of numbers and letters on the horizontal axis represents the combinations of elevation and disturbance level. The numbers 1, 2, 3, 4, and 5 represent gradients 1, 2, 3, 4, and 5, respectively (see study design for each elevation gradients); L, M, and H represent low, moderate, and high disturbance, respectively.

**Figure 7 insects-15-00669-f007:**
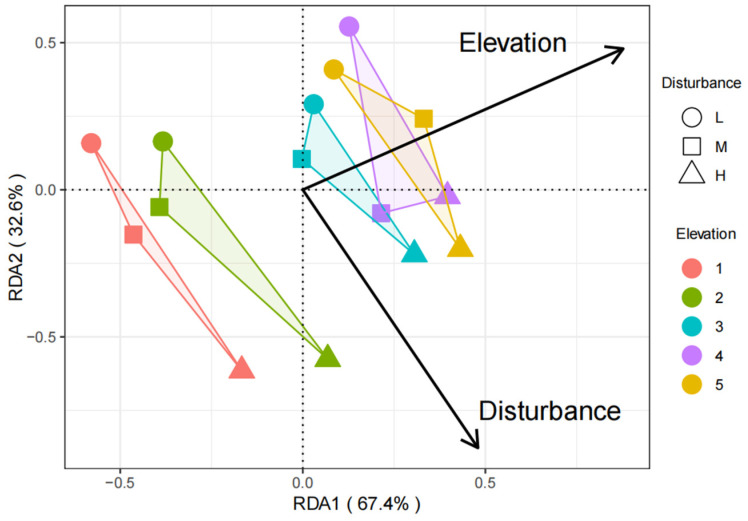
Redundancy analysis of insect communities at different elevation gradients and disturbance levels. Note: L, M, and S represent low, moderate, and high disturbance, respectively. The numbers 1, 2, 3, 4, and 5 represent elevation gradients 1, 2, 3, 4, and 5, respectively (see study design for each elevation ranges).

**Table 1 insects-15-00669-t001:** The composition of the insect community in the studied area.

Order	Number of Families	Proportion	Number of Individuals	Proportion
Orthoptera	5	6.25%	427	8.30%
Hemiptera	20	25.00%	1735	33.75%
Coleoptera	22	27.50%	2015	39.19%
Diptera	10	12.50%	250	4.86%
Hymenoptera	6	7.50%	531	10.33%
Neuroptera	4	5.00%	48	0.93%
Lepidoptera	6	7.50%	21	0.41%
Trichoptera	1	1.25%	50	0.97%
Phasmida	1	1.25%	5	0.10%
Dermaptera	1	1.25%	10	0.19%
Mantodea	1	1.25%	10	0.19%
Odonata	2	2.50%	5	0.10%
Blattaria	1	1.25%	4	0.08%
Total	80	100%	5141	100%

**Table 2 insects-15-00669-t002:** Multi-model reasoning based on elevation and human disturbance.

Models	AIC	R^2^
Model 1: EL	353.68	0.15
Model 2: HD	360.45	0.11
Model 3: EL + HD	330.69	0.31
Model 4 *: EL × HD	328.92	0.33

Note: EL, elevation; HD, human disturbance. The best model is indicated with an asterisk (*). Because the ΔAIC for models 3 and 4 is less than 2, we conducted model averaging analyses. The results confirm that model 4 is the optimal model.

## Data Availability

The data presented in this study are available on request from the corresponding author.

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
