# Peer review of "Elevation and Human Disturbance Interactively Influence the Patterns of Insect Diversity on the Southeastern Periphery of the Tibetan Plateau"

_insects, 2024, doi:10.3390/insects15090669_

Round 1
Reviewer 1 Report
Comments and Suggestions for Authors
Dear authors: Congratulations on an interesting and well written piece of work. Please consider my comments to methods regarding name of parameter of diversity under q = 1 and q = 2, in my opinion they reflect but are not the same as Shannon's or Simpson's indices, for the sake of clarity; also please specify the meaning of the acronym AIC for a statistical procedure. If possible, be sure to mention when morphospecies (number of morphospecies) were used (e.g., when number of species is mentioned), same for supplementary material, if you consider it appropriate these numbers are not given, only abundance by family. Some other minor suggestions are marked in the text. Finally, if possible, please shorten discussion a little, emphasizing your results, less emphasis on results from other studies and general discussion.

Comments on the Quality of English LanguageSeveral minor English suggestions are marked to the text, some are rather small (e.g., use of singular for a general usage of a term).
Author Response
Dear authors: Congratulations on an interesting and well written piece of work. Please consider my comments to methods regarding name of parameter of diversity under q = 1 and q = 2, in my opinion they reflect but are not the same as Shannon's or Simpson's indices, for the sake of clarity;
Response: Thank you for this professional suggestion. We fully agree that methods regarding name of parameter of diversity under q = 1 or 2 can reflect but are not the same as Shannon's or Simpson's indices. In the data analysis of the manuscript, we employed direct calculation of the absolute value of the diversity index. Our previous statement may have caused some confusion, so we have refined the description in the revised manuscript. Please see revised manuscript in L197-199.
also please specify the meaning of the acronym AIC for a statistical procedure.
Response: Thank you for this important comment. We have added the specific meaning of AIC where it is first mentioned, please see revised manuscript in L217-218.
If possible, be sure to mention when morphospecies (number of morphospecies) were used (e.g., when number of species is mentioned), same for supplementary material, if you consider it appropriate these numbers are not given, only abundance by family.
Response: Thank you for your valuable suggestion. We fully agree with the reviewer’s opinion. Based on the existence of a large number of unnamed insect species, we used morphospecies for identification. Since there are still a large number of unnamed species at the species and genus level, we analyzed the diversity of insects based on the family diversity to ensure better identification accuracy, and the foundation of using the family as an analytical category has already been verified (Zou et al 2020), as described in L188-192 in manuscript. Following your suggestion, we have checked and revised the manuscript to ensure that the description of identification category and the analytical category were consistent.
Some other minor suggestions are marked in the text.
Response: Thank you for your careful review of our manuscript. We have made all required revisions in our revised manuscript. You may find all the revisions in tracked changes.
Finally, if possible, please shorten discussion a little, emphasizing your results, less emphasis on results from other studies and general discussion.
Response: Thank you for this important comment. Strictly following your suggestion, we have condensed our discussion section accordingly. Please see revised manuscript in L344, L352。
Reviewer 2 Report
Comments and Suggestions for Authors
This paper is well designed and very well written. The topic is of general interest and the analyses sound. Therefore I have very minor comments.
The main one is that given that the study aimed at collecting all types of insects, and that some insect orders need special methods of capture, some groups like Blattodea and Odonata are clearly underrepresented in the samples. There are some conclusions about these orders which are based in so few specimens that are not justified. My suggestion is to remove them.
Other comments are included in the attached pdf.

Author Response
This paper is well designed and very well written. The topic is of general interest and the analyses sound. Therefore I have very minor comments. The main one is that given that the study aimed at collecting all types of insects, and that some insect orders need special methods of capture, some groups like Blattodea and Odonata are clearly underrepresented in the samples. There are some conclusions about these orders which are based in so few specimens that are not justified. My suggestion is to remove them.
Response: Thank you very much for your positive evaluation on our manuscript. We fully agree with the reviewer's suggestion and have removed the content recommended by the reviewer in the revised manuscript. Please see revised manuscript in L31, L295, L408.
Other comments are included in the attached pdf.
Response: Thanks you for your meticulous review of this manuscript. Following your suggestion, we have revised our manuscript accordingly. You may find all revisions in tracked changes.
Reviewer 3 Report
Comments and Suggestions for Authors
The elevation pattern of diversity is a hot topic in current research. When discussing this pattern, the influence of human disturbance has often been considered. For example, under a unimodal model, harsh environmental conditions typically lead to a monotonic decrease in diversity. However, this pattern becomes unimodal due to significant human disturbance at lower elevations, resulting in a decrease in diversity.
Therefore, in the introduction, one paragraph should clearly address the elevation pattern of diversity without considering human disturbance. The subsequent paragraph should then discuss the influence of human disturbance on this pattern, which would align well with the topics the authors are exploring.
Additionally, I am uncertain whether the intermediate disturbance hypothesis applies to human activity, especially when human disturbance generally leads to homogeneous environmental conditions.
Importantly, this study is based on field surveys rather than experiments. The general collinearity between elevation and human disturbance (with higher levels of disturbance at lower elevations) does exist. However, in Figure 7, the study seems to account for all combinations and demonstrates that collinearity does not exist in their case. I checked the criteria used to determine different levels of human disturbance, and they merely cited a reference without showing a detailed table explaining how each site was assigned different levels. The reference [34 or maybe 44 by the citation order] they cited seems not to provide the criteria.
It is crucial that the study accounts for all combinations and shows low collinearity, and the study design should be emphasized to show their novelty and contributions.
Otherwise, the collinearity issue means that a one-way ANOVA cannot be used to compare diversity across different elevations or levels of human disturbance. Regarding model selection, if the AIC difference is smaller than 4 or 6, the models are considered similar. While considering those similar models, the authors can still note that the interaction between elevation and human disturbance exists. However, they need to address the issue of collinearity in the discussion.
Considering these issues, the authors may need to rewrite the entire manuscript. Thus, I refrain from commenting on the discussion section for now, but I am willing to review the next version before the final evaluation.
Author Response
The elevation pattern of diversity is a hot topic in current research. When discussing this pattern, the influence of human disturbance has often been considered. For example, under a unimodal model, harsh environmental conditions typically lead to a monotonic decrease in diversity. However, this pattern becomes unimodal due to significant human disturbance at lower elevations, resulting in a decrease in diversity.
Therefore, in the introduction, one paragraph should clearly address the elevation pattern of diversity without considering human disturbance. The subsequent paragraph should then discuss the influence of human disturbance on this pattern, which would align well with the topics the authors are exploring.
Response: Thank you for your valuable suggestion. Following your suggestion, in paragraphs 2 and 3 of the revised manuscript, we discussed relevant studies on the effects of elevation and human disturbances on biodiversity when these factors act independently. However, in paragraph 4, due to limited available articles focusing on interactions between elevation and human disturbances, our discussion in this section is relatively brief. We have further discussed these interactions in the discussion section based on our obtained study results.
Additionally, I am uncertain whether the intermediate disturbance hypothesis applies to human activity, especially when human disturbance generally leads to homogeneous environmental conditions.
Response: Thank you for this professional comment. The intermediate disturbance hypothesis is defined as“the highest diversity is maintained at intermediate scales of disturbance”. However, the explicit explanation regarding the applicability of this hypothesis to human activity is lacking, despite human disturbances often being considered and discussed as a contributing factor in studies involving other taxa (eg:Mayor, S., Cahill, J., He, F. et al. Regional boreal biodiversity peaks at intermediate human disturbance. Nat Commun , 2012, 3(1142). Therefore, we also included the corresponding discussion as a possible explanation of the results.
Importantly, this study is based on field surveys rather than experiments. The general collinearity between elevation and human disturbance (with higher levels of disturbance at lower elevations) does exist. However, in Figure 7, the study seems to account for all combinations and demonstrates that collinearity does not exist in their case. I checked the criteria used to determine different levels of human disturbance, and they merely cited a reference without showing a detailed table explaining how each site was assigned different levels. The reference [34 or maybe 44 by the citation order] they cited seems not to provide the criteria.
Response: Thank you for your professional suggestion. Indeed, the possible collinearity between elevation and human disturbance may exist. In considering of this, in the design of the study, we distribute the sampling sites with different levels of disturbances in each elevation as evenly as possible, although it is not possible to completely equalize the distribution of disturbance levels in each elevation gradient due to the actual conditions in the field, this design is expected to reduce the collinearity between elevation and disturbance. Besides, we also performed statistical tests to check for collinearity: the Pearson correlation coefficient between elevation and human disturbance was -0.20, with p value was 0.172, indicating that the correlation between the two was low and statistically insignificant; the variance inflation factor (VIF) in the model is 1.03, also indicate that the collinearity is negligible.
We fully agree that our previous description on the specific classification criteria of human disturbances and the references cited might be insufficient, to address this issue, we have revised the description of the classification of disturbances and updated the references. Please see revised manuscript in L170.
It is crucial that the study accounts for all combinations and shows low collinearity, and the study design should be emphasized to show their novelty and contributions.
Otherwise, the collinearity issue means that a one-way ANOVA cannot be used to compare diversity across different elevations or levels of human disturbance. Regarding model selection, if the AIC difference is smaller than 4 or 6, the models are considered similar. While considering those similar models, the authors can still note that the interaction between elevation and human disturbance exists. However, they need to address the issue of collinearity in the discussion.
Considering these issues, the authors may need to rewrite the entire manuscript. Thus, I refrain from commenting on the discussion section for now, but I am willing to review the next version before the final evaluation.
Response: Thank you for your valuable comment. As mentioned in our previous response, all statistical tests results indicate insignificant collinearity, therefore one-way ANOVA can be used under the current situation. Furthermore, the difference in AIC values between the additive model and the interaction model was found to be less than 4, suggesting statistical similarity between these two models. To better assess their fitting effectiveness, we conducted a model averaging analysis by assigning weights to each model according to their respective AIC values, the results from this analysis indicated that the interaction model outperformed the additive model in explaining the effects of elevation and disturbance on insect diversity. Following your suggestion, we have emphasized novelty and contributions in the study design section as well as included a discussion on collinearity in the discussion section. Please see L170-173 and L367-370 for revisions.
Round 2
Reviewer 3 Report
Comments and Suggestions for Authors
I still have a few comments and suggestions. Please provide the range of the IDI for each level of disturbance. Please include the observation points in Figure 5. Additionally, in the Introduction and/or Discussion, please show the detailed linkage between the IDI (human disturbances) and environmental heterogeneity in your specific case.
Author Response
Comments: I still have a few comments and suggestions. Please provide the range of the IDI for each level of disturbance. Please include the observation points in Figure 5. Additionally, in the Introduction and/or Discussion, please show the detailed linkage between the IDI (human disturbances) and environmental heterogeneity in your specific case.
Response: Thank you for your professional comments. Following your suggestion, we have revised our manuscript accordingly. First, we have incorporated the IDI range for each level of disturbance, please see in lines 174-176. Additionally, we have added observation points in Figure 5, as illustrated in the revised manuscript. Furthermore, we have established a detailed linkage between human disturbances and environmental heterogeneity within in the introduction and discussion in the revised manuscript, you may find revisions in lines 118-123, 384-385.
Round 3
Reviewer 3 Report
Comments and Suggestions for Authors
The statement about the linkage between human disturbances and environmental heterogeneity is too general. Why does the linkage exist, and how does it manifest? You could provide specific examples from your study area to illustrate this connection.
Author Response
Comments: The statement about the linkage between human disturbances and environmental heterogeneity is too general. Why does the linkage exist, and how does it manifest? You could provide specific examples from your study area to illustrate this connection.
Response: Thank you for your professional suggestion. We believe our manuscript can be further improved under your help. Following your suggestion, we have detailed the statement about the linkage between human disturbances and environmental heterogeneity, and we have incorporated specific examples from our study to illustrate the connection. You may find the revisions in lines 121-124 and 369-401.